# Fell on Black Days: Analyzing the Song Lyrics of Chris Cornell for Insight into Depression and Suicide

**DOI:** 10.3390/ijerph20166621

**Published:** 2023-08-21

**Authors:** Kevin P. Conway, Patrick McGrain, Michelle Theodory

**Affiliations:** 1Genetic Epidemiology Research Branch, National Institute of Mental Health, Bethesda, MD 20892, USA; 2Department of Criminal Justice, Gwynedd Mercy University, Gwynedd Valley, PA 19437, USA; mcgrain.p@gmercyu.edu; 3Department of Biomedical Informatics, Harvard University, Boston, MA 02115, USA; mtheodory@hms.harvard.edu

**Keywords:** depressive disorder, suicide, addiction, qualitative analysis, natural language processing

## Abstract

Chris Cornell was a guitarist, singer, songwriter, and pioneer of grunge music. Cornell struggled with mental illness and addiction and incorporated these themes into his song lyrics. At age 52, Cornell died by suicide in his hotel bathroom following a live performance. This mixed-methods study examines Cornell’s song lyrics for references to negative words and themes related to depression and suicide. Two coders independently reviewed lyrical transcripts to identify the primary theme, secondary theme(s), and valence (positive or negative). Sentiment analysis, a natural language processing technique, was used to examine word frequency and valence. Songs (N = 215) were predominantly (79%) negative and contained more negative (N = 3244, 56.1%) than positive (N = 2537, 43.9%) words. Thematic analysis by stage of career shows a narrowing focus on depression, failed relationships, and morbid thoughts. Themes of depressed mood, death, and suicide were common and increased by stage of career. By applying qualitative and quantitative techniques to song lyrics, this study revealed that Cornell’s songs reflect a narrative of negativity consistent with someone experiencing depression and thoughts of death and suicide. Like personal notes and poems, song lyrics may reflect symptoms of depression and suicidal thoughts warranting clinical attention.

## 1. Introduction

Suicide is a leading cause of mortality in the United States. In 2020, suicide was the second or third leading cause of death among adolescents and young adults (ages 10–34) and the fourth among adults aged 35–44 [1]. Further, the national age-adjusted rate of suicide has increased notably over the past two decades for both males and females [2].

Prior research has identified myriad risk factors for suicide, as summarized in recent reviews [3,4]. Background risk factors include demographics (e.g., age, sex, race), geographic location, culture, seasonal variation, physical illness, and sleep disturbances. Psychosocial factors include relationship difficulties, problematic substance use, recent crises, social isolation, adverse childhood experiences, and easy access to lethal means [2]. Mental health plays a major role in suicide risk, as most cases of death by suicide involve a person with a history of a mental disorder [5]. Mood disorders and substance use disorders are markedly large risk factors, especially when comorbid [6]. Despite abundant research identifying risk factors for suicide, it remains the case that many suicides occur without warning, highlighting the need to identify novel sources of information about suicide risk, especially among at-risk individuals.

Research suggests that artists are at increased risk of mental disorders, notably mood disorders and substance use disorders, as well as suicide [7,8,9,10,11,12,13]. The risk may be especially elevated among artists who express themselves through the written word [8,14]. Studies of the works of famous poets who have died by suicide offer some insight into the internal world of writers who died by suicide, revealing common themes of mental illness, depressed mood, thoughts about death, interpersonal difficulties, unrelenting pain, and a sense of hopelessness/helplessness [8,12,15,16,17]. Although the potential causes of suicide are numerous and heterogeneous, one hypothesis is that artists are at elevated risk due to underlying psychiatric vulnerability aggravated by factors related to artistic professions including public displays or performances, frequent rejection of highly personal creations, and intense criticism from self and others [13,15].

Despite artistic similarities between poetry and songwriting, scant research has examined musical lyrics as a source of information regarding mental illness and suicide. Tuft et al. [18] examined the song lyrics of Ian Curtis, the frontman of post-punk band Joy Division who died by suicide in 1980; several song references described Curtis’ struggles with epilepsy, depression, and interpersonal strain. In an analysis of song lyrics by Black Sabbath, the pioneers of heavy metal music, Conway and McGrain [19] found that references to drug use and addiction were common and shifted over time from positive to negative, a pattern consistent with natural-history studies and neurobiological models of addiction [20,21]. Sule and Inkster [22] inspected the lyrics of “Stan,” one of Eminem’s most popular hip-hop songs, for references to mental illness and identified references to personality disorder, depression, suicidal ideation, and addiction. The existing literature is further limited by methodology, as all prior studies exclusively used only qualitative analyses. Despite the growth in the use of natural language processing (NLP) techniques to scan text for indicators of mental illness, few studies analyze narrative writings [23], and none have applied NLP techniques to analyze musical lyrics.

Christopher John Cornell was a guitarist, singer, and songwriter who, with fellow Seattle-based musicians including Kurt Cobain of Nirvana and Eddie Vedder of Pearl Jam, pioneered and popularized grunge music in the late 1980s and early 1990s. Grunge represented an alternative subgenre that fused punk rock with heavy metal, incorporated powerful guitar distortion and feedback, and featured intense vocals that embraced melancholic and anguished lyrics. Cornell was the lead singer and primary lyricist for three groups—Soundgarden, Temple of the Dog, and Audioslave—and a solo musician. Recognized by music critics and listeners as one of the greatest voices in the history of rock, Cornell sold more than 30 million albums, won three Grammy awards, and inspired numerous posthumous tributes and memorials, including a bronze statue installed outside Seattle’s Museum of Popular Culture. Cornell’s songs often incorporated themes including mental illness and drug addiction, which he personally experienced [24,25]. At age 52, Cornell was found dead in his hotel bathroom shortly after performing with Soundgarden at Detroit’s Fox Theatre on May 18, 2017. His death was ruled a suicide (by hanging) by the Wayne County Medical Examiner’s Office.

We are aware of only one scientific paper to examine the lyrics of Cornell. Intrigued by the timing of Cornell’s suicide in the season (spring) when most suicides occur [26], Schwartz [27] examined Cornell’s lyrics from several songs to suggest that Cornell suffered from depression with mixed features, a clinical presentation associated with increased risk of suicide [28,29]. Building upon Schwartz [27] and incorporating NLP, the current mixed-methods study examined all musical lyrics written by Cornell for references to depression and suicide. The study poses two research questions (RQs). RQ1: To what extent do Cornell’s song lyrics use words that are negative or reflect depressive or suicidal themes? RQ2: To what extent does the use of negative words or themes vary by stage of career?

## 2. Materials and Methods

This study analyzed all songs containing lyrics written and recorded by Cornell. The study employed a mixed-methods design involving qualitative and quantitative approaches consistent with prior research in this area [19,30,31,32]. These approaches were supplemented using NLP, as described below. Lyrics were obtained from online databases of song lyrics.

For the qualitative approach, we developed a semi-structured coding scheme that was applied to each song. Two coders independently examined written lyrical transcripts utilizing 1st-level and 2nd-level codes to systematically evaluate each song and determine their primary theme, secondary theme(s), and valence. The codes were identifed after several reviews of the lyrics as well as guidance from the literature on factors associated with mental health, especially depression and suicidal ideation.

The 1st-level code (primary theme) represented the overarching idea or meaning conveyed by a song’s lyrics. Five primary themes emerged: mood, relationships, political, drugs, and lifestyle. Each song had one primary theme. Songs that expressed positive (e.g., happy, optimistic) or negative (e.g., sad, gloomy, depressive) mood states were coded as “mood.” Songs expressing love and loss within the context of personal relationships were coded as “relationships.” Songs about a political topic, stance, or commentary were coded as “political.” Songs focusing on drugs or drug use (e.g., consumption, getting high) were coded as “drugs.” Songs describing everyday existence, life choices, and career-related issues were coded as “lifestyle”.

The next step involved identifying the 2nd-level codes (secondary theme). Because many songs were multidimensional in nature, each song was coded with secondary themes that supplemented, clarified, or contextualized the primary theme. Each song had several secondary themes. Six secondary themes emerged: addiction, love/loss, professional, fate/destiny, depression, and other. Songs referencing pathologic drug use (e.g., loss of control, negative consequences) were coded as “addiction.” Songs about the ups and downs of interpersonal relationships were coded as “love/loss.” Songs about employment and career experiences were coded as “professional.” Songs expressing a sense of a future beyond one’s control were coded as “fate/destiny.” Songs describing a negative mood state were coded as “depression.” Finally, songs that referenced other themes were coded as “other”.

Valence identified songs as positive or negative. Songs with a preponderance of optimistic and positivistic language, imagery, and subjects (e.g., goodness, happiness, love, joy, light) were coded as positive. Songs with primarily negative language, imagery, and subjects (e.g., despair, hopelessness, darkness, dread) were coded as negative.

All songs were coded by two authors (KC and PM), first independently and then jointly, to reach preliminary consensus on all codes. During this process, the coders took extensive notes to identify key imagery, references, and specific lyrics exemplifying the essence of each song.

The quantitative methods involved sentiment analysis to complement the qualitiative approach. Sentiment analysis is an NLP technique that scans text for information (e.g., emotions, attitudes) and subsequently assigns a sentiment value to that text. The current study used “sentimentr” [33] because, unlike other approaches, it considers the role of negators and valence shifters within text to improve classification. Referring to a dictionary lexicon, positive and negative words were initialy polarized with a +1 or −1, respectively. Words surrounding each polarized word were then considered as negators or valence shifters, defaulting to four words before and two words after the polarized word. For example, although the word “happy” could be positive, preceding the word with “not” alters the meaning of the word; in this case, the word would be classifed as negative. Finally, for each song, the values of the words were summed and divided by the square root of the word count to result in an overall sentiment score. Using 0 as a threshold (indicating neutral), songs with a value higher than 0 were classified as positive and below 0 as negative.

Word clouds were generated to visually represent the frequencies of positive and negative terms in the catalog of song lyrics. To do so, we used the extract_sentiment_terms() function to extract a list of positive, neutral, and negative words per song. The attributes() function was used to produce an overall list containing every word used across all songs, along with a polarity score assigned to each word (as described above). Two separate word clouds were used to represent the frequencies of positive and negative terms in all song lyrics, including repeated terms, with a maximum of 200 words displayed per figure. More frequently occurring words appear as larger font sizes compared to less frequently used words. Counts of specific words related to morbid themes (death or dying, the afterlife, and suicide) were also conducted.

Variables were entered into a database that was supplemented with information about each song derived from various discographic sources (album sleeve, Wikipedia, etc.), including the song title, album name, track number, song duration, and title and year of album release. Instrumentals (N = 7), lyrical songs not written by Cornell (N = 11), and different versions of the same song (N = 5) were excluded. This resulted in a total of 215 lyrical songs available for analysis.

The frequency of the songs by themes and valence were examined overall and by stage of career. Stage of career was defined chronologically according to discography and included 7 stages: Early Soundgarden (N = 82), Temple of the Dog (N = 10), Early Solo (N = 14), Audioslave (N = 41), Middle Solo (N = 35), Later Soundgarden (N = 15), and Final Solo (N = 18).

## 3. Results

Of the 215 songs, most (N = 148, 69%) were written when Cornell was frontman for Soundgarden, Temple of the Dog, and Audioslave. The remaining 67 songs (31%) were written as a solo artist. The most common primary theme was mood (N = 91, 42%), followed by relationships (N = 56, 26%), politics (N = 31, 14%), drugs (N = 26, 12%), and lifestyle (N = 11, 5%). The most common secondary themes were depression (N = 58, 27%), other (N = 52, 24%), love/loss (N = 50, 23%), addiction (N = 20, 9%), and professional (N = 14, 7%).

The coders and “sentimentr” agreed on the valence for 148 songs (69%). Cases of disagreement between were flagged and reevaluated by the coders, independently at first and then jointly, to reach consensus. In all cases of disagreement (N = 67), the coders classified the song as negative whereas “sentimentr” classified the song as positive. For these cases, the coders revisited their codes and reclassified 13 (19%) songs from negative to positive (in agreement with “sentimentr”). This resulted in a final agreement rate of 75%.

RQ1: To what extent do Cornell’s song lyrics use words that are negative or reflect depressive or suicidal themes?

Most songs were classified as negative whether by coders (90%) or “sentimentr” (59%). Following the reconciliation process described above, there were 180 (84%) songs classified as negative in the final analysis. Further, the negativity permeated all primary themes. Figure 1 shows that most songs were negative across the primary themes of mood (76 of 91, 84%), relationships (40 of 56, 71%), politics (30 of 31, 97%), drugs (24 of 26, 92%), and lifestyle (10 of 11, 91%). Of note, the two most common primary themes (mood and relationships), accounting for 70% of all songs, were predominantly (79%) negative.

The songs contained more negative words (N = 3244, 56%) than positive words (N = 2537, 44%) (see Figure 2). From a thematic perspective, many of the negative words reflect depressive thoughts and imagery including loneliness (e.g., “alone,” “leave,” “lost”), suffering (e.g., “broken,” “break,” “pain”), death (e.g., “drown,” “dead,” “dying”), darkness (e.g., “black,” “dark,” “cold”), and sadness (e.g., “cry”, “crying”, “tears”). The most common negative word (“wrong”) evokes a powerful sense of injustice. Regarding the positive words, specific words such as “love”, “good”, and “sun” connote happiness, suggesting that the author is able to find joy in some aspects of his life. Additionally, words such as “truth” and “right” suggest that his outlook was not entirely morose, and that he saw positive things occurring in the world. The figure also shows that many of the most common negative words occurred more frequently than the most common positive words, as indicated by the larger font size for many negative words. 

Explicit mentions of death or dying appeared in 63 (29%) songs. The word “dead” (or “deadly”) occurred 49 times across 23 songs; “dying” occurred 25 times across 15 songs; “death” occurred 20 times across 10 songs; “grave” (or “graveyard,” “gravestone,” or “tomb”) occurred 17 times across 15 songs. In general, these words were used throughout the discography to suggest that life is nothing more than a slow process of dying interrupted with painful experiences. In “Searching with My Good Eye Closed,” life is described as a mental struggle from its inception (“Stop you’re trying to bruise my mind, I can do it on my own. Stop you’re trying to kill my time, it’s been my death since I was born.”). In “Like a Stone,” thoughts of death preoccupy Cornell’s mind (“On a cobweb afternoon in a room full of emptiness. By a freeway, I confess I was lost in the pages of a book full of death, reading how we’ll die alone”). In “The Day I Tried to Live,” this sentiment is especially explicit (“Words you say never seem to live up to the ones inside your head. The lives we make never seem to ever get us ’nowhere but dead.”). “Blind Dogs,” a homage to the late Jim Carroll, a counterculture extraordinaire known for his autobiographical book “The Basketball Diaries,” rejects religion as a respite from the misery of life (“Dead on my feet while my nightmare walks. I fell asleep where the freeway talks. Suffer to swim and dying to sink.”). “Tighter and Tighter” endorses heroin use for both its euphoric effect and its deadly potential (“Warm and sweet swinging from a window’s ledge. Tight and deep, one last sin before I’m dead, before I’m dead. A sucking holy wind will take me from this bed tonight. And bloody wits another hits me and I have to say goodbye.”)

References to states of existence after death appeared in 22 (10%) songs. Specifically, the word “heaven” occurred 51 times across 12 songs; “hell” occurred 33 times across 10 songs. Thematically, the references suggest that an afterlife provides happiness and freedom from pain and suffering. In “Our Time in the Universe,” Cornell describes the certainty of death (“save the dying arms of midnight … beneath miles of earth and sky”) as an opportunity to forever reunite with a long-lost loved one (“it’s our time in the universe, yours and mine”). “Black Hole Sun” describes a desire to escape from despair, possibly by injecting heroin (“black hole sun won’t you come and wash away the rain”) to manufacture paradise (“heaven send hell away”). “The Last Remaining Light” depicts an existence defined by omnipresent pain (”through your spine, and every nerve, where I watch, and I wait, and yield to the hurt”), pessimism (“and if you don’t believe the sun will rise, stand alone and greet the coming night, the last remaining light”), and a longing for relief that only an afterlife can provide (”heaven waits, for those who run down your winter, and underneath your waves, where you watch and you wait, and pray for the day”). In “Say Hello to Heaven,” a tribute to the overdose death of friend and fellow musician Andrew Wood, Cornell explicitly pleads for an exit (“please, mother mercy, take me from this place”) from the path followed thus far (“you better seek out another road, ‘cause this one has ended abrupt”) and welcomes an afterlife with open arms (“say hello to heaven, heaven, yeah). “When Bad Does Good,” Cornell’s final song, describes the certainty of death (“standing beside an open grave your fate decided, your life erased, your final hour has come today”) as intentional action (“I choose a side and I will show no pity”), to relieve mental anguish (“the fire of your temples burning”) that is a net-positive panacea (“let it be understood, sometimes bad can do some good”).

Explicit mentions of suicide were less common, occurring 11 times across 4 (2%) songs. In “Like Suicide,” the word is used paradoxically to describe a relationship that is both loving and tragically self-destructive (“Love’s like suicide, dazed out in a garden bed. With a broken neck lays my broken gift, just like suicide”). Elsewhere, “suicide” is used more concretely to describe the hazards of road trips in “Get on the Snake” (“get on the snake with a suicide machine”) and hazardous driving in “Watch Out” (“She hit the corners with her foot on the floor. Watch out! She likes the Lincoln with the suicide doors”). Reflecting a more contemplative tone, “Silence the Voices” ruminates over humanity’s inability to heed opposing views on issues of grave significance (e.g., whether to go to war), especially when decisions lead to self-inflicted demise (“a man in endless suicide”).

Aside from explicit mentions, suicide was coded as a secondary or tertiary theme in several songs. “No Such Thing” describes not only how depression can negate even the most positive experiences (“I saw the world, it was beautiful. But the rain got in and ruined it all”), but also how persistent negative cognitions can worsen a depressed state (“And I thought too hard on the world, and as usual I slumped too far into the void”). Above all, the song contemplates suicide (“So what gives me the right to think that I could throw away a life? Even mine”) as a final solution to silence the pain (“But my finger’s on the trigger and I’ll turn off the world”). Suicide is also alluded to repeatedly to overcome grief from a failed relationship. “Cleaning My Gun” describes losing the love of his life (“I believed that you would be my only one … But somewhere in the ashes of this burning lovers’ game. Somehow you decided that you would find another flame”) will result in unbearable remorse (“And as you lay sleeping with your eyes softly shut. I’ll be cleaning my gun, cleaning my gun …. When heaven or hell takes this life, I’ll be done.”). This subject repeats in “Worried Moon,” an anxious song about how the pain of rejection (“And I hope but I don’t know If she will have me back again. Or only want me for a friend. And leave a stain across my heart. That never washes out.”) leads to thoughts about suicide (“Yeah if it all goes wrong and I’m a heart without a home. Maybe you can talk me out of doing myself in.”)

RQ2: To what extent does the use of negative words or depressive or suicidal themes vary over stage of career?

Songs were predominantly negative across Cornell’s career. Figure 3 shows that, during Early Soundgarden (1987–1996), nearly all songs were negative (79 of 83, 95%). All 10 songs written with Temple of the Dog (1991) were negative. More than half of the songs written for the two albums with Audioslave (2005–2006) were negative (25 of 41, 61%). Similarly, all but two songs written during Later Soundgarden (2012–2014) were negative (13 of 15, 87%). Approximately three quarters of the songs during Cornell’s solo career were negative including Early Solo (11 of 15, 73%), Middle Solo (29 of 35, 83%), and Final Solo (14 of 18, 78%). Songs were predominantly negative whether written as a group (126 of 145, 87%) or as a solo artist (54 of 68, or 79%).

The distribution of the five primary themes varied by stage of career (see Figure 4). During Early Soundgarden, the songs covered all themes in a well-distributed pattern. Temple of the Dog focused predominantly on drugs (50%) and politics (30%), plus one song each on relationships and mood. Early Solo and Middle Solo concentrated similarly on relationships (39%) and mood (35%), followed by drugs (20%). Audioslave and Later Soundgarden focused on mood (61%) followed by politics (16%) and relationships (18%). Final Solo concentrated almost exclusively on relationships (50%) and mood (44%). The songs increasingly focused on relationships or mood, especially after Temple of the Dog, until peaking in Final Solo (94%).

The distribution of secondary themes also varied by stage of career (see Figure 5). The top panel shows that early Soundgarden evidenced all six secondary themes. Temple of the Dog focused disproportionally on addiction relative to each of the three other themes on the album. Although the remaining stages included a diversity of themes, especially during Audioslave, a focus on depression and love/loss emerged during Early Solo and increased thereafter. The bottom panel shows that the proportion of songs coded as either depression or love/loss increased steadily until peaking in Final Solo (at 94%).

Finally, explicit mentions of death, the afterlife, and suicide varied by stage of career. Songs referencing suicide were rarer during Early Soundgarden (N = 1) than Middle Solo (N = 3) and Final Solo (N = 3). The pattern by stage of career was similar for songs with words referencing death (“dead,” “deadly,” “dying,” “death,” “tomb,” “grave,” “graveyard,” “gravestone”), the afterlife (“heaven,” “hell”), and “suicide.” Therefore, we combined these specific words into a category called morbid words, which appeared in 100 (47%) songs. Overall, the proportion of songs with morbid words increased by stage of career. The proportion was lowest for Early Solo (21%) and highest for Later Soundgarden (100%), and moderate for other stages (40–50%).

## 4. Discussion

This mixed-methods study examined words and themes related to depression and suicide in all song lyrics written by musician Chris Cornell, a pioneer of grunge music, who died by suicide at the age of 52. Results show that Cornell’s songs are predominantly negative as measured by valence, negative/positive word count, and theme. The dominance of negativity was observed by both qualitative and quantitative methods, across all primary themes, and when Cornell was both part of a group and a solo artist. Thematically, Cornell’s songs focused primarily on depressed mood, failed relationships, political frustration, addiction, and morbid thoughts. As depicted in “Fell on Black Days,” Cornell appears to accept depression as an inescapable fate.

The pattern of negativity over Cornell’s career varied by study measure. On the one hand, the proportion of songs classified as negative decreased over time. It was highest during Early Soundgarden and Temple of the Dog, lowest during Early Solo and Audioslave, and moderate during Middle Solo, Later Soundgarden, and Later Solo. At no point, however, was the proportion of negative songs lower than 61%. On the other hand, the analysis of the primary themes by stage of career suggests a narrowing focus on the predominantly negative themes of mood and relationships at the expense of other themes. This is most striking when contrasting the primary themes in Early Soundgarden to Final Solo. As shown in Figure 4, whereas 60% of the songs during Early Soundgarden were coded as mood or relationships, 91% of the songs in Final Solo were coded as such. Analysis of the secondary themes by stage of career further specifies an increased focus on depression and love/loss, both of which are predominantly negative.

Cornell’s songs were not only overwhelmingly negative, but they often included words that conveyed thoughts and feelings of a depressive and suicidal nature. Throughout the discography, the dominant primary theme was mood, and the prevailing secondary theme was depression. Songs with direct or indirect references to suicide were not uncommon and were written predominately as a solo artist and later in his career. Further, the proportion of songs with mentions of morbid thoughts—death, an afterlife, and suicide—generally increased over time. Qualitatively, the thematic focus on such thoughts intensified throughout Cornell’s career, suggesting a worsening mental state of an individual in need of intervention. Indeed, Cornell’s final album, “Higher Truth,” is seemingly one unambiguous statement of misery and longing for relief from suffering. Songs such as “Nearly Forgot My Broken Heart,” “Dead Wishes,” and “Worried Moon’’ conjure feelings of depression, anxiety, loneliness, and a deeply cynical view that life is devoid of hope and fulfillment. In “Before We Disappear,” Cornell sings about depression, death, sorrow, and the finality of life. He sings about hope and being unlovable, a painful dichotomy he is at a loss to explain. In “Through the Window,” he is prophetic, wishing he could give his loved ones back the happiness that is gone. Overall, the album deals with themes of melancholy, fate, depression, and emotional pain, with a single love song (“Only These Words”) written to his daughter. In this final chapter of his discography, Cornell both accepts and expresses a personal truth that he no longer wishes to live a life defined by suffering.

Results from the current study are similar to the scant few other lyrical-analysis studies. In the case of post-punk frontman Ian Curtis of Joy Division, his lyrics and live performances personified struggles with epilepsy, depression, and interpersonal strain [18]. The analysis of the lyrics of heavy-metal pioneer Black Sabbath revealed a longitudinal shift in the valence of substance-use references from positive to negative that closely corresponded with increasingly pathological drug use by the band members [19]. Schwartz [27] identified several Cornell songs containing lyrics that support the idea that Cornell suffered from depression with mixed features, a clinical presentation that may increase the risk of suicide [28,29]. Common across these reports is a narrative of personal suffering expressed through lyrics that helps authenticate the songwriters as artists and connect with audiences. Unfortunately, the seriousness of the personal suffering expressed in lyrics tends to be romanticized, under-recognized, or mismanaged by others [18,34,35,36]. As Cornell writes autobiographically in “Outshined,” appearances that exude self-confidence belie an inner world of misery and self-loathing (“*I’m looking California and feeling Minnesota*”). Our results also align with studies of the works of famous poets who have died by suicide, which collectively reveal themes of mental illness, depressed mood, morbid thoughts, pain, and hopelessness/helplessness [8,16,17]. It is paradoxical and tragic that factors contributing to these artists’ success may have also contributed to their demise [18,34].

Certainly, the current analysis does not suggest that Cornell’s song lyrics portend his death by suicide. However, when considered in conjunction with other suicide risk factors attributed to Cornell—male, white, middle age, current or prior mental illness (depression, anxiety, addiction), early adversity, interpersonal conflict, etc. [24,25,27]—his writings about death and suicide carry extra weight. These risk factors may be aggravated by unique stressors of artistic careers including frequent rejection of personal creations [13,15], as well as the demands of a touring musician (e.g., frequent onstage performances, excessive travel, circadian disruption). Aside from depression, Cornell suffered from social anxiety and was taking Lorazepam at the time of his death; both social anxiety [37,38] and benzodiazepines [39] are associated with increased risk of suicide.

More research is needed on the distal and proximal factors that may contribute to mental health and suicide, especially among artists and other at-risk groups. While the work of poets has long been explored for literary analysis, newer modalities including social media could allow researchers to take a broader scan of mental health in a community. Researchers are increasingly applying NLP techniques to social media platforms to detect depression and suicide, revealing opportunities and challenges [23]. Techniques that flag abberations in text communications in near real-time (e.g., alerts to social-media postings reflecting sudden changes in mood, cognition, or behavior) may be promising, especially within the context of distal risk factors such as history of mental illness and impulsivity. Additionally, given the link between physical activity and suicidal ideation [40] future studies should also consider incoporating objective measures of physical activity to identify atypical patterns. Ecological studies using Ecological Momentary Assessment and other mobile technologies to measure core domains (e.g., mood, substance use, suicidal thoughts, sleep, physical activity) may be especially helpful in uncovering associations among risk factors and their direction of influence [41,42,43].

These findings also have implications for NLP and underscore the value of research that incorporates both quantitative and qualitative approaches. The lower rate of songs classified as negative by sentiment analyses may suggest a positive bias of computational approaches that rely on word-valence scores. Although NLP and other techniques offer a less burdensome and potentially more efficient approach to manual qualitative analysis, these advantages must be evaluated against bias and other potential threats to validity. Because words likely have different valences/meanings in songs than in everyday language, it is important for computational approaches to consider the role of context (e.g., lyrics vs. every-day conversation) and form of expression (e.g., poem vs. factual statement), as well as the value of ongoing evaluation and revisions by humans.

This study has some limitations. Although the use of natural language processing provides some degree of objectivity, bias and subjectivity of human coding is unavoidable. The study assumes that Cornell’s lyrics reflect his personal experiences; while this assumption is supported by published work [24,25,27], it is open to interpretation. Relatedly, the lyrics are highly personal and may not be generalized beyond this songwriter. The case-study approach has inherent drawbacks that may introduce confirmatory bias. The retrospective design cannot be used for predictive purposes. The study is subject to potential confounding due to unmeasured factors. Limitations notwithstanding, this study offers an innovative and potentially informative approach to gain insight into the inner world of someone experiencing depression and thoughts of death and suicide.

## 5. Conclusions

Chris Cornell’s song lyrics were predominately negative and increasingly focused on depression, failed relationships, and morbid thoughts. Themes of depressed mood, death, and suicide were common and increased by stage of career. Like personal notes and poems, song lyrics may reflect depression and suicidal thinking warranting clinical attention, especially in the presence of known risk factors for suicide.

## Figures and Tables

**Figure 1 ijerph-20-06621-f001:**
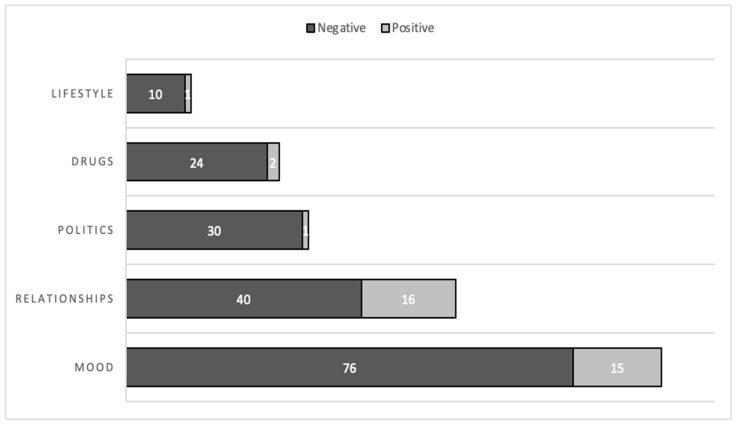
Number of Songs Coded as Negative and Positive, by Primary Theme.

**Figure 2 ijerph-20-06621-f002:**
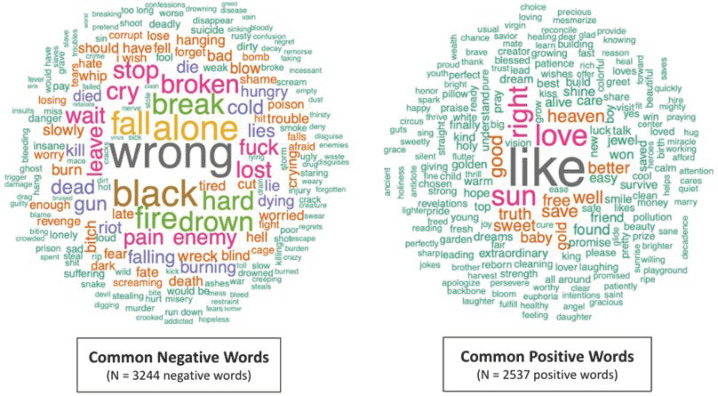
Word Clouds Depicting Common Negative and Positive Words in the Song Lyrics of Chris Cornell.

**Figure 3 ijerph-20-06621-f003:**
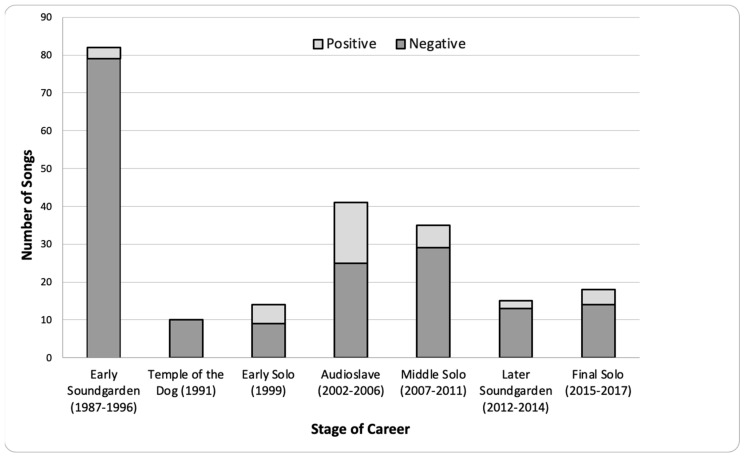
Song Valence by Stage of Career.

**Figure 4 ijerph-20-06621-f004:**
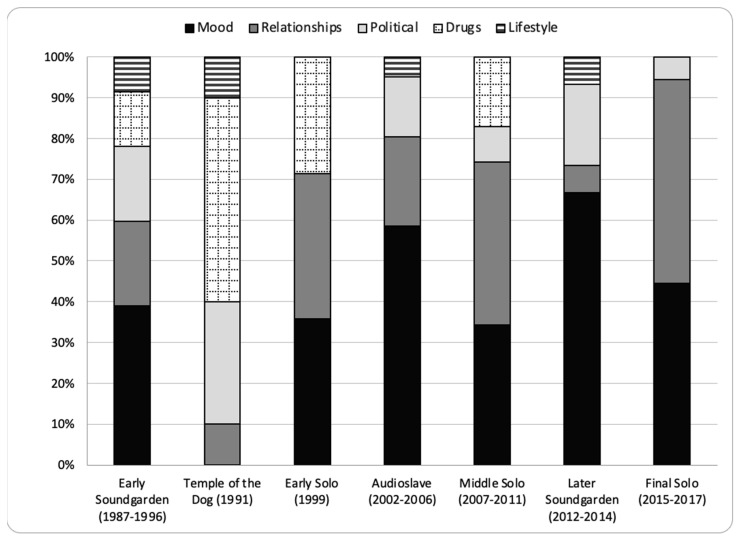
Primary Theme by Stage of Career.

**Figure 5 ijerph-20-06621-f005:**
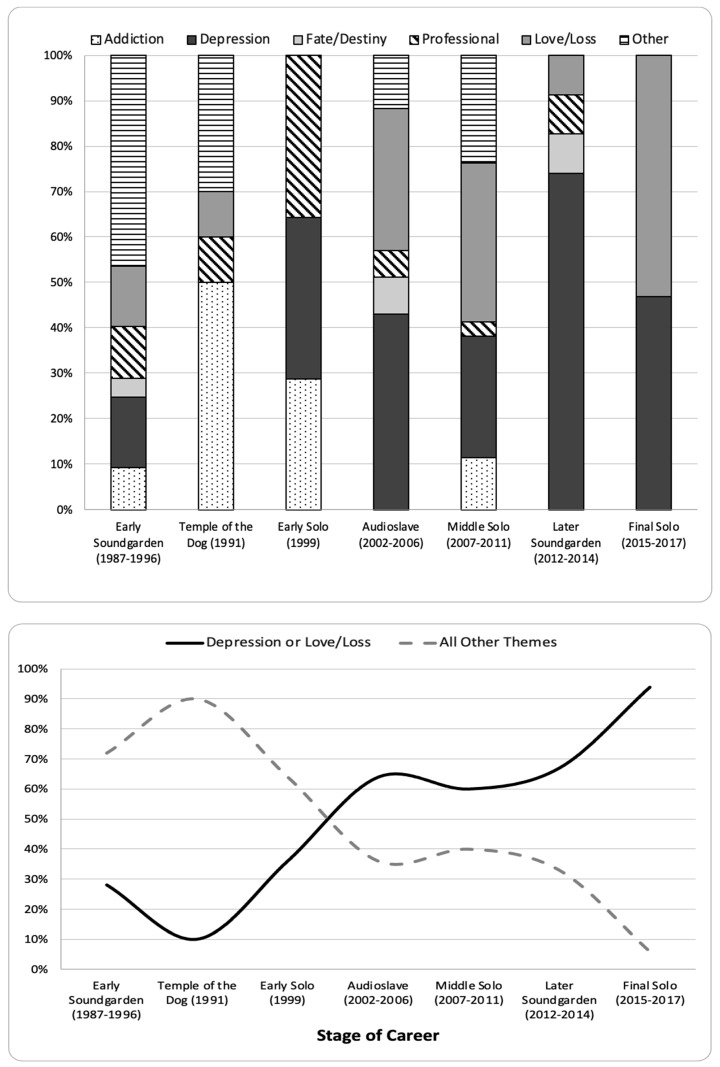
Secondary Theme by Stage of Career: All Themes (Top Panel) and Depression or Love/Loss versus All Other Themes (Bottom Panel).

## Data Availability

Data sharing complies with the 2023 National Institutes of Health Policy for Data Management and Sharing. Data used for this study will be shared via the generalist repository Dryad, which provides metadata, persistent identifiers (i.e., Digital Object Identifier), and long-term access. Data will be made available in Dryad at the time of the associated publication. Information needed to make use of the data (e.g., variable names, definitions, codes, information about missing data, metadata, etc.) will be available in codebooks and shared alongside final datasets.

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
