# Peer review of "Fell on Black Days: Analyzing the Song Lyrics of Chris Cornell for Insight into Depression and Suicide"

_ijerph, 2023, doi:10.3390/ijerph20166621_

Round 1

Reviewer 1 Report

Dear authors,

thank you for your article. It is innovative and interesting. Only small things could be improved:

1. The use of NLP and sentimentr for computer-based, automatic identification of lyrics content is difficult to understand for those unfamiliar with it. A clearer explanation would be helpful.

2. Methods: In line 174, it would be helpful to say that lyrics that were not "at least partially" written by Cornell were excluded. Otherwise this fact would be unclear until the results are presented in line 177.

3. The colors of the graphics only make sense in color printouts. Printed in black and white, it is impossible to distinguish between blue and red, making it difficult to read. A mix of dark and pale colors would be helpful.

4. The proportions of negative and positive words do not vary greatly at 56.1 to 43.9%. A more detailed description of the positive words to compare with the negative words described from line 206 below would be a good comparison to weight both types.

5. In line 286 you say that "most songs" from Audioslave were negative. With only 61% negative lyrics I see it more as an outweighing part.

6. The visual presentation is very helpful, but there are a lot of graphics. A reduction could be made here to point out most important information, for example by only describing panel B in Figures 3 - 5 in the text. Alternatively, these panels would be interesting in comparison to positive topics instead of "all other topics" in order to be able to weight them again.

Author Response

We appreciate the thoughtful review and suggestions for change.  Below, we enumerate each Reviewer’s comments and our responses to them.  In the manuscript, all relevant changes are highlighted in yellow. 

Reviewer 1

Dear authors, thank you for your article. It is innovative and interesting. Only small things could be improved:

  1. The use of NLP and sentimentr for computer-based, automatic identification of lyrics content is difficult to understand for those unfamiliar with it. A clearer explanation would be helpful.

    Response: The description of NLP and sentiment have been clarified to make it more digestible to a wide audience.

  1. Methods: In line 174, it would be helpful to say that lyrics that were not "at least partially" written by Cornell were excluded. Otherwise this fact would be unclear until the results are presented in line 177.

    Response: This has been clarified in the Methods. 

  1. The colors of the graphics only make sense in color printouts. Printed in black and white, it is impossible to distinguish between blue and red, making it difficult to read. A mix of dark and pale colors would be helpful.

    Response: The figures have been revised in black and white. 

  1. The proportions of negative and positive words do not vary greatly at 56.1 to 43.9%. A more detailed description of the positive words to compare with the negative words described from line 206 below would be a good comparison to weight both types.

    Response: We have included additional text to describe some of the common positive words. 

  1. In line 286 you say that "most songs" from Audioslave were negative. With only 61% negative lyrics I see it more as an outweighing part.

    Response: We have revised this language as follows.  “More than half of the songs written for the two albums with Audioslave (2005-2006) were negative (25 of 41, 61%).”

  1. The visual presentation is very helpful, but there are a lot of graphics. A reduction could be made here to point out most important information, for example by only describing panel B in Figures 3 - 5 in the text. Alternatively, these panels would be interesting in comparison to positive topics instead of "all other topics" in order to be able to weight them again.

    Response: Thank you for this suggestion.  We have eliminated Panel B in Figures 3-5.  Instead, we describe the results from these panels in the text.  Figure 5 retains Panel C (now Panel B) because it clearly shows that the songs increasingly focused on depression or love/loss. As a result of these changes, the revised figures include only the information that is best displayed graphically. 

Reviewer 2 Report

Thank you for your work, which is really interesting.  It is well known that the arts, and especially music, are powerful means for people to project and express the most intimate thoughts and feelings, and the case study you present is a clear example.  I hope this kind of research can be done at a larger scale.  If through people's and artists' creative output some suicides can be foreseen and prevented, it would be a great achievement.

In the material and methods sections, lines 104-122 are not very clear to me, especially the distinction between primary and secondary themes.  It would be wonderful if you could clarify further how you developed/decided the coding scheme that was applied to each song. I would recommend that you add a table that include the two levels of coding and the connection between them.

At the discussion and conclusion section, it would be recommended that you add how you would move forward with this topic in future research studies. This could inspire other researchers interested in the same topic.

Author Response

We appreciate the thoughtful review and suggestions for change.  Below, we enumerate each Reviewer’s comments and our responses to them.  In the manuscript, all relevant changes are highlighted in yellow.  

Reviewer 2

Thank you for your work, which is really interesting.  It is well known that the arts, and especially music, are powerful means for people to project and express the most intimate thoughts and feelings, and the case study you present is a clear example.  I hope this kind of research can be done at a larger scale.  If through people's and artists' creative output some suicides can be foreseen and prevented, it would be a great achievement.

  1. In the material and methods sections, lines 104-122 are not very clear to me, especially the distinction between primary and secondary themes. It would be wonderful if you could clarify further how you developed/decided the coding scheme that was applied to each song. I would recommend that you add a table that include the two levels of coding and the connection between them.

    Response: This section has been revised. We have included additional text to better describe the coding scheme applied to each song.

  2. At the discussion and conclusion section, it would be recommended that you add how you would move forward with this topic in future research studies. This could inspire other researchers interested in the same topic.

    Response: We have included a few statements about future directions in the Discussion.  While we encourage additional text-mining research on this topic, we also emphasize the incorporation of supplemental measures collected simultaneously and in real-world settings. 

    We also included a new reference (40) and reordered the subsequent accordingly.